# COVID-19-Associated Pulmonary Aspergillosis in Intensive Care Unit Patients from Poland

**DOI:** 10.3390/jof9060666

**Published:** 2023-06-13

**Authors:** Magdalena Skóra, Mateusz Gajda, Magdalena Namysł, Jerzy Wordliczek, Joanna Zorska, Piotr Piekiełko, Barbara Żółtowska, Paweł Krzyściak, Piotr B. Heczko, Jadwiga Wójkowska-Mach

**Affiliations:** 1Chair of Microbiology, Faculty of Medicine, Jagiellonian University Medical College, Czysta 18 Street, 31-121 Krakow, Poland; mateusz14.gajda@uj.edu.pl (M.G.); pawel.krzysciak@uj.edu.pl (P.K.); piotr.heczko@uj.edu.pl (P.B.H.); jadwiga.wojkowska-mach@uj.edu.pl (J.W.-M.); 2Department of Microbiology, University Hospital in Krakow, Macieja Jakubowskiego 2 Street, 30-688 Krakow, Poland; mnamysl@su.krakow.pl; 3Interdisciplinary Intensive Care Clinic, Jagiellonian University Medical College, Macieja Jakubowskiego 2 Street, 30-688 Krakow, Poland; j.wordliczek@uj.edu.pl; 4Center for Innovative Medical Education, Jagiellonian University Medical College, Medyczna 7 Street, 30-688 Krakow, Poland; jzorska@su.krakow.pl; 5Intensive Care Unit, University Hospital in Krakow, Macieja Jakubowskiego 2 Street, 30-688 Krakow, Poland; 6Department of Internal Diseases and Circulatory Failure, Center of Pulmonology and Thoracic Surgery in Bystra, Juliana Fałata 2 Street, 43-360 Bystra, Poland; ppiekielko@szpitalbystra.pl; 7Department of Pulmonology and Respiratory Failure, Center of Pulmonology and Thoracic Surgery in Bystra, Juliana Fałata 2 Street, 43-360 Bystra, Poland; 8Center for Innovative Therapy, Clinical Research Coordination Center, University Hospital in Krakow, Macieja Jakubowskiego 2 Street, 30-688 Krakow, Poland; bzoltowska@su.krakow.pl

**Keywords:** *Aspergillus*, COVID-associated pulmonary aspergillosis, CAPA, invasive aspergillosis, SARS-CoV-2

## Abstract

Coronavirus disease 2019 (COVID-19) has been shown to be a favoring factor for aspergillosis, especially in a severe course requiring admission to the intensive care unit (ICU). The aim of the study was to assess the morbidity of CAPA among ICU patients in Poland and to analyze applied diagnostic and therapeutic procedures. Medical documentation of patients hospitalized at the temporary COVID-19 dedicated ICU of the University Hospital in Krakow, Poland, from May 2021 to January 2022 was analyzed. In the analyzed period, 17 cases of CAPA were reported with an incidence density rate of 9 per 10 000 patient days and an incidence rate of 1%. *Aspergillus fumigatus* and *Aspergillus niger* were isolated from lower respiratory samples. Antifungal therapy was administered to 9 patients (52.9%). Seven patients (77.8%) received voriconazole. The CAPA fatality case rate was 76.5%. The results of the study indicate the need to increase the awareness of medical staff about the possibility of fungal co-infections in ICU patients with COVID-19 and to use the available diagnostic and therapeutic tools more effectively.

## 1. Introduction

Invasive aspergillosis is one of the most common life-threatening fungal infections, affecting more than one million people all over the world, mainly patients with prolonged neutropenia, hematologic malignancies, recipients of allogenic stem cell transplants or solid organ transplants, patients with prolonged use of corticosteroids, and also those with inherited and congenital severe immunodeficiencies [1,2]. Invasive aspergillosis also occurs in critically ill patients in the absence of immunological host factors [3,4]. Recently, COVID-19 has been shown to be a predisposing factor for mycosis caused by *Aspergillus*. Pulmonary aspergillosis has been reported to be a relatively common co-infection in severe COVID-19 patients requiring admission to the intensive care unit (ICU) and has been related to increased mortality [5,6,7,8,9]. The phenomenon of increased susceptibility of patients suffering from acute viral pneumonia to invasive aspergillosis has been reported before in the case of influenza [3,10,11]. The pathomechanism of these co-infections is not fully understood, but it is most likely related to damage to the epithelium of the respiratory tract as a result of primary viral infection, which promotes *Aspergillus* to invade tissue, as well as dysregulation of immune response due to infection and the use of some drugs, i.e., steroids [12,13,14].

One of the reasons for the high mortality rate in COVID-19-associated aspergillosis is difficulties in diagnosis of the fungal infection and, consequently, the lack of or delayed implementation of antifungal therapy. Limitations in the detection of aspergillosis result from nonspecific clinical symptoms and radiographic findings, which may be indistinguishable from the coronavirus disease alone. A key problem in the diagnostics of pulmonary aspergillosis is to distinguish tissue invasion from transient *Aspergillus* colonization or even contamination of the sample with environmental mycobiota. The criteria allowing the diagnosis of invasive fungal disease with varying levels of certainty developed by the European Organization for Research and Treatment of Cancer (EORTC) and the Mycoses Study Group Education and Research Consortium (MSGERC) are difficult to apply to patients hospitalized in intensive care units, including patients with coronavirus disease [2]. Most of these patients are immunocompetent and therefore do not meet the fundamental criterion of probable and possible aspergillosis, which is immunosuppression. The definition of proven aspergillosis requires histopathologic evidence of mycotic infection from primarily sterile material, which may not be available to collect from critically ill patients. Even before the COVID-19 pandemic, the separate criteria for the diagnosis of invasive pulmonary aspergillosis (IPA) in ICU patients were developed—the AspICU algorithm [15]. This algorithm was aimed to discriminate *Aspergillus* colonization from putative IPA in critically ill patients. In 2021, a new version, the BM-AspICU algorithm, was developed [16]. It has been dedicated to any patient requiring ICU admission for respiratory distress, regardless of risk factors, and was extended by *Aspergillus* biomarkers.

Due to problematic diagnosis of aspergillosis in patients with severe COVID-19, the European Confederation for Medical Mycology (ECMM) and the International Society for Human and Animal Mycology (ISHAM) prepared special guidelines providing the definitions of proven, probable, and possible COVID-associated pulmonary aspergillosis (CAPA), which includes recommendations for the diagnosis and treatment of CAPA [17]. This guideline is helpful in interpreting laboratory findings, but the diagnosis of CAPA still remains a great challenge. In Poland, one of the important factors influencing this situation is the lack of awareness of the medical staff about fungal superinfection in patients with coronavirus disease and the lack of implementation of appropriate diagnostic and therapeutic procedures.

The aim of the study was to assess the morbidity of CAPA among ICU patients hospitalized in the temporary COVID-19-dedicated ICU of the University Hospital in Krakow (UHK) and to analyze applied diagnostic and therapeutic procedures. The secondary objective was to determine the CAPA fatality case rate and to assess the impact of selected factors on mortality.

## 2. Materials and Methods

The retrospective study included 1614 patients (18,810 patient days, pds) with confirmed COVID-19 with SARS-CoV-2 PCR tests (COBAS 6800, Roche or in the CITO mode GeneXpert System, Cepheid, USA) treated in the temporary COVID-19-dedicated ICU of UHK from 1 May 2021 to 31 January 2022. Due to instability, not all patients had computed tomography (CT) scans, and a bedside X-ray was performed in most cases. Samples from the lower respiratory tract were collected for mycological culture from 214 (13.3%) patients. GM- and BGD-detecting assays were performed in 35 (2.2%) and 37 (2.3%) patients with 48 and 51 tests, respectively. All patients who underwent serological diagnostics also had a culture of samples from the lower respiratory tract. *Aspergillus* fungi were isolated from 16 patients (7.5% of patients who underwent mycological culture from the lower respiratory tract). One patient had a positive result for GM antigen without positive *Aspergillus* culture. These 17 cases were retrospectively recognized as CAPA and underwent a detailed analysis.

The morbidity of CAPA was calculated as (1) incidence rate in relation to the emergence of new CAPA cases in the population of ICU patients over a study period and (2) incidence density rate using as denominator the summed patient days of observation [18].

*Aspergillus* fungi were isolated from BAL obtained in bronchoscopy procedure or other lower respiratory tract samples: bronchial lavage, tracheal aspirate, and secretion from the bronchial tree, called, for the purposes of this publication, non-bronchoscopic lavage (NBL). The materials were cultured on Sabouraud glucose agar with gentamicin and chloramphenicol (OXOID) and incubated at 25 °C and 35 °C for 7 days. Species identification was based on the microscopic examination of preparations stained with Lactophenol Cotton Blue (Merck KGaA) and evaluation of the protein profile using matrix-associated laser desorption ionization time-of-flight mass spectrometry (MALDI-TOF MS) VITEK^®^MS (bioMerieux, ver. V3.2). Antifungal susceptibility testing was performed by the E-test method (Liofilchem, Roseto degli Abruzzi, Italy) on RPMI-1640 agar medium (Liofilchem, Roseto degli Abruzzi, Italy). Minimal inhibitory concentration (MIC) values were read after 48 h incubation at 35 °C. Interpretation of MICs for antifungal agents was based on breakpoints determined by the European Committee on Antimicrobial Susceptibility Testing [19]. Serological tests detecting GM antigen (Platelia Aspergillus Ag, Bio-Rad, Marnes-la-Coquette, France), mannan (M) antigen (Platelia Candida Ag Plus, Bio-Rad, Marnes-la-Coquette, France), and 1, 3-β-D-glucan (BDG) antigen (Fungitell, Associates of Cape Cod, Inc., East Falmouth, MA, USA) were performed in accordance with the manufacturers’ instructions.

CAPA was defined according to the 2020 ECMM/ISHAM consensus criteria with three grades: proven, probable, and possible, depending on mycological evidence [17]. Proven CAPA was diagnosed when histopathological or direct microscopic examination of pulmonary tissue showed fungal elements morphologically resembling *Aspergillus* spp. with invasive growth into tissue and tissue damage, or in the case of isolation of *Aspergillus* from a pulmonary tissue sample. Probable CAPA was applied to patients with radiological findings, i.e., pulmonary infiltrate or cavitating lesion, not attributed to another cause, and if one of the following criteria was met: microscopic detection of fungal elements in BAL, positive *Aspergillus* culture from BAL, serum GM index > 0.5, BAL GM index ≥ 1.0. Patients with possible CAPA presented radiological criteria and at least one mycological feature: fungal elements indicating a mold present in non-bronchoscopic lavage, *Aspergillus* culture from non-bronchoscopic lavage, single non-bronchoscopic lavage GM index > 4.5, non-bronchoscopic lavage GM index > 1.2 twice or more [17]. As a second diagnostic criterion for pulmonary aspergillosis, a BM-AspICU clinical algorithm was used to distinguish probable IPA from possible IPA or *Aspergillus* colonization [16]. The entry criteria were: positive *Aspergillus* culture from the lower respiratory tract or clinical sings or radiological signs. Next, the presence of host factors associated with immunosuppression and other risk factors was analyzed, as well as mycological evidence (BAL and NBAL culture results, GM detection in BAL and plasma/serum).

The data analysis was performed using the R programming language (4.3.0 21 April 2023) and RStudio (v 2023.03.1) as the statistical software. Given the non-parametric nature and the small number of data, two specific tests, namely the Fisher’s exact test and the Mann–Whitney U test, were employed. Statistical significance was assessed using a predetermined threshold of *p* ≥ 0.05. This work was approved by the Bioethics Committee of Jagiellonian University (approval no. 1072.6120.353.2020 from 16 December 2020). All data analyzed during this study were anonymized prior to analysis.

## 3. Results

The incidence density rate of COVID-associated pulmonary aspergillosis (CAPA) was 9 per 10,000 pds, and the incidence rate was 1.0%. According to the 2020 ECMM/ISHAM consensus criteria [17], 4 cases (23.5%) were classified as probable and 13 cases (76.5%) as possible. Based on the BM-AspICU algorithm [16], 4 cases (23.5%) were also identified as probable IPA, and the remaining 13 (76.5%) as possible IPA or *Aspergillus* colonization. In two cases, the two diagnostic criteria used were inconsistent (Table 1). Most of the patients with CAPA (*n* = 10, 58.8%) were male. The average age was 63.7 years old (range 33–78 years old). The most common comorbidities were hypertension (*n* = 6, 35.3%), then heart failure (*n* = 3, 17.6%). Two patients (11.8%) had previous pulmonary disease—asthma or chronic obstructive pulmonary disease (COPD)—and one (5.9%) had a history of pulmonary neoplasm (Table 2 and Appendix A). All of the patients were under invasive mechanical ventilation, and most (*n* = 15, 88.2%) were intubated on the first day of hospitalization. With one exception, all of the patients (*n* = 16, 94.1%) were on corticosteroid therapy on the first or second day of hospitalization in ICU (Table 2 and Appendix A). Bacterial pathogens from the lower respiratory tract were isolated from 76% of CAPA patients (*n* = 13), and 71% (*n* = 12) had bacteremia (Table 2 and Appendix A). Almost all patients were receiving antibacterial drugs (*n* = 16, 94.1%) (Table 2 and Appendix A). Carbapenems were administered in each case, while linezolid was used in 10 patients (58.9%), vancomycin in 5 cases (29.4%), and other beta-lactams in 8 cases (47.0%). Two patients (11.8%) received piperacillin with tazobactam. During hospitalization, 15 patients (88.2%) were on combination antibiotic therapy with at least two drugs. Only one patient (5.9%) received tocilizumab. The mean blood pH of the patients was in the lower limit of normal values (7.36) with an average arterial partial pressure of CO_2_ of 52.29 mmHg and arterial partial pressure of O_2_ of 74.06 mmHg. The average respiratory rate was defined as 28.6/min with a blood saturation with oxygen at a level of 93%. During single days of hospitalization, the respiratory rate setting exceeded 35/min. The CAPA patients were mechanically ventilated for 72.1% of their hospitalization time in the ICU. All CAPA patients had changes characteristic of SARS-CoV-2 infection described as multifocal ground-glass opacities of rounded morphology with or without consolidation or visible intralobular lines on X-ray or CT examinations. Only 3 patients (17.7%) had altered lymph nodes. The detailed clinical and radiological characteristics of the patients are presented in Appendix A.

In 14 cases with a positive *Aspergillus* culture from the lower respiratory tract (87.5%), these fungi were isolated from non-bronchoscopic lavage samples only in 2 cases (12.5%) from BAL. In 7 cases (43.8%), samples were collected on the day of admission to the ICU, and in 9 (56.2%), during hospitalization in ICU. On average, samples from the lower respiratory tract for mycological cultures were collected after 5 days (median 4 days, range 0–21 days) from the start of ICU stay. Results of the cultures for *Aspergillus* were obtained after 4 to 31 days of hospitalization in the ICU (average 11 days, median 11 days) and after 3 to 11 days (average 6 days, median 7 days) after the collection of clinical material from the lower respiratory tract. *Aspergillus fumigatus* was identified in 14 patients (87.5%) and *Aspergillus niger* in 2 (12.5%). In addition to *Aspergillus*, fungi from the genus *Candida* were isolated from all patients from the same clinical samples, most often (81.2%) *Candida albicans*. Other species isolated from individual patients were identified as *Candida dubliniensis*, *Candida inconspicua*, *Candida krusei*, *Candida tropicalis*, and *Penicillium* spp. *A. fumigatus* strains for which antifungal susceptibility testing was performed were susceptible to amphotericin B, itraconazole, and voriconazole (Figure 1 and Appendix A).

In 8 patients (50%) with positive *Aspergillus* culture, GM antigen in serum was tested, in 2 (12.5%) additionally in respiratory samples—1 in BAL and 1 in non-bronchoscopic lavage. These were single tests, except for one patient who was tested twice one week apart. One patient from whom *Aspergillus* was isolated from non-bronchoscopic material (6.3%) had a positive GM (index 7.6). In two patients (12.5%), GM was detected in respiratory samples, in one patient in BAL with a GM index of 0.85, but it was not found in serum, and one in non-bronchoscopic lavage with a GM index of 3.85 and also with a negative result for a serum sample. Seven patients (43.8%) were tested for serum BDG, and all the tests were positive, with BDG concentrations above 80 pg/mL. One patient was diagnosed as CAPA only based on the results of serological tests: positive GM antigen (GM index 8.48) and positive BDG antigen (523.45 pg/mL) in a single serum sample, with negative culture, result for *Aspergillus* (Appendix A).

Blood cultures were performed for all 17 patients with CAPA at least once during hospitalization in the ICU. Only one patient (5.9%) had a positive culture for fungi; *C. albicans* was isolated from blood samples taken through the central catheter and from arterial blood. *Aspergillus* species were not cultured (Appendix A).

Antifungals were administered to 9 of the 17 patients (52.9%). Most of the patients (*n* = 7, 77.8%) received voriconazole. Four patients (23.5%) received fluconazole prior to *Aspergillus*-positive culture, and in two cases, voriconazole was administered after obtaining a positive *Aspergillus* culture from lower respiratory tract material. One patient died, and the second, two weeks after voriconazole administration, received caspofungin. The therapy turned out to be effective, and the patient stayed alive. Five patients with antifungal therapy (55.5%) received only voriconazole after positive *Aspergillus* culture from respiratory samples. In two patients, the drug was administered on the same day as the culture result, which was 7 days after the collection of clinical material, and the patients died the day after or two days after starting therapy. In one case, voriconazole was started the day after *Aspergillus* isolation, and the patient stayed alive. In the other two cases, the drug was started two and three days after the positive *Aspergillus* culture. The first patient died, and the second stayed alive in the ICU but died about 2 months after transfer to the palliative care ward. Due to the large time span between *Aspergillus* isolation and death, this case was not included in the calculation of the CAPA fatality rate. Eight CAPA patients (47.0%) did not receive any antifungal therapy, including two patients whose mycological results indicated probable CAPA, and 7 of them (87.5%) died. Summing up, among 17 patients who had probable or possible CAPA, as many as 14 died (2 patients after being transferred to other UHK units). Thus, after excluding patients who died about 2 months after *Aspergillus* isolation and transfer to the palliative care ward, the CAPA fatality case rate was 76.5%.

## 4. Discussion

So far, research on invasive aspergillosis has focused primarily on patients who are most often affected by this infection, i.e., patients with severe immune disorders with neutropenia, such as neoplastic diseases of the hematopoietic and lymphoid tissues and transplant recipients [22,23,24]. The infection has also been known to occur in COPD patients [25,26]. Among the described risk factors for the development of invasive pulmonary aspergillosis are hospitalization in the ICU and severe influenza virus-related pneumonia [3,4]. Recently, viral–fungal co-infections with *Aspergillus*, *Mucorales*, and *Candida*, including *C. auris*, have emerged during the SARS-CoV-2 virus pandemic [27]. Co-infections have not been reported to be common in COVID-19 patients, but their occurrence was found to be much more frequent in patients with coronavirus disease requiring ICU treatment [28,29].

The main objective of the study was to assess the epidemiology of pulmonary aspergillosis in patients who required hospitalization in the ICU due to severe SARS-CoV-2 infection in a large hospital in Poland and to analyze the diagnostic and therapeutic methods used in invasive aspergillosis in our CAPA group. During the 9-month retrospective analysis, a total of 17 cases of CAPA were diagnosed. The CAPA incidence density rate was 9 per 10,000 pds (or incidence rate 1%), which is lower than the lowest rates reported in other studies where it ranged from 1.6% to 38% (median 20.1%) [27,30]. Such a large discrepancy between different studies was mainly due to the various criteria used for the diagnosis of CAPA, which resulted in some *Aspergillus* colonization cases being classified as aspergillosis or vice versa. It is also directly related to the availability of diagnostic methods used in invasive aspergillosis. The lack of proven CAPA cases and the low percentage of probable cases (25%) with a simultaneously high percentage of deaths (76.5%) in our analysis suggest that the frequency of CAPA in our study may be underestimated. In our opinion, the lack of BAL sampling and infrequent testing for GM antigens is one of the reasons for these results. This may also indicate problems with the diagnosis of invasive mycoses in general, not just aspergillosis, and could be related to the physicians’ approach to fungal infections, especially the use of available diagnostic tools in everyday clinical practice in critically ill patients. To the best of the authors’ knowledge, this is the first analysis of this type in Poland; therefore, we are unable to compare the results obtained with other data from Poland.

Diagnosis of invasive mycoses in patients infected with SARS-CoV-2 is a particularly big challenge due to the limiting of methods that pose a risk to subjects or medical staff, especially in obtaining valuable samples of biological materials, which are lung tissue and BAL. In our study, in any event, the diagnosis of the proven CAPA could not be made following the ECMM/ISHAM criteria [17]. The main reason was the lack of histopathological material, which was not collected at the high risk of complications in critically ill patients. Alternatively, an important assessment tool remains a flexible bronchoscopy (FB), which provides direct visualization of the tracheobronchial tree and sampling for bacteriological and mycologic examinations. As an aerosol-generating procedure, at the beginning of a pandemic, it was considered only a last resort in emergency situations [31]. In subsequent waves of coronavirus infection, new evidence has shown that the resignation of direct endoscopic evaluation significantly impairs the diagnosis of respiratory diseases, especially bacterial and fungal co-infections [32,33]. BAL, obtained during FB, is one of the main samples for the diagnosis of respiratory infections with a lower diagnostic threshold than other lower respiratory tract materials (tracheal aspirate, non-bronchoscopic lavage, sputum) [17,34]. It should be noted that all patients with acute respiratory failure requiring mechanical ventilation and high-percentage oxygen supplementation are exposed to the hazard of a drop in oxygenation during the FB. The probability of other most common complications, such as hemodynamic instability, bronchospasm, cardiac arrhythmias, hypotension, bleeding, hemoptysis, myocardial infarction, and death, also remains important in this group of patients [34,35]. According to the authors’ experience, non-invasive diagnostic methods, such as endotracheal aspirate for collecting an appropriate respiratory sample, dominate in Polish ICU in ventilator-associated pneumonia cases. In our study, only two patients with CAPA (11.8%) had *Aspergillus* isolated from BAL samples. In other cases, materials obtained without endoscopy were used—endotracheal aspirate and non-bronchoscopic lavage, which are easier to obtain in intubated patients but less representative of the lower respiratory tract and not fully validated for fungal biomarkers detection [17]. A definite predominance of non-bronchoscopic samples caused greater difficulties in the interpretation of culture results and limited the performance of GM tests, which, according to the manufacturer’s recommendations, should be determined in serum or BAL.

The simultaneous use of various diagnostic methods increases the chances of detecting aspergillosis. In accordance with ECMM/ISHAM guidelines [17], the following mycological evidence could be used to diagnose CAPA: histopathological or direct microscopic detection of fungal hyphae in pulmonary samples, positive *Aspergillus* culture from pulmonary samples, positive *Aspergillus* PCR from pulmonary samples, and GM detection, not only in serum and BAL materials but also in other lower respiratory tract samples, with a different cut-off index depending on the biological material. In our hospital, the dominant laboratory technique for the detection of aspergillosis is culture. Microscopic evaluation of the clinical material, which is a valuable examination accompanying culture, was not performed in respiratory samples from COVID-19 patients for safety reasons. It is known that the clinical importance of *Aspergillus* in the respiratory tract in critically ill patients is difficult to assess and has been discussed long before the COVID-19 pandemic [15,36]. The wide distribution of *Aspergillus* in nature and inhalation of spores into the respiratory tract creates difficulties in interpreting the results of mycological diagnosis based on culture. *Aspergillus* fungi are isolated from respiratory samples from approximately 0.7–7% of patients hospitalized in the ICU, and COPD, corticosteroid therapy, and neutropenia are described as the major predisposing factors for the occurrence of *Aspergillus* in the respiratory tract of those patients [15,37]. However, the presence of fungi in the respiratory tract is not tantamount to fungal disease and requires careful analysis. The criteria for the diagnosis of invasive mycoses, which have been developed mainly for immunosuppressed patients, are difficult to apply to ICU patients. The reasons are several: the lack of possibility to collect samples of reliable biological materials, i.e., lung tissue biopsies, the lack of classic host factors that favor fungal infection, i.e., immunosuppression, and nonspecific radiological findings caused by mechanical ventilation. In addition, in patients with primary viral infections of the lower respiratory tract, the clinical picture is blurred by the overlapping of various etiological factors, and the results of imaging tests are difficult to interpret unambiguously.

In UHK, apart from culture, the only methods used to diagnose invasive aspergillosis are GM detection by ELISA and BDG testing, which is a nonspecific fungal marker, and its use is currently not recommended for the diagnosis of aspergillosis or is marginally supported [2,17,38]. *Aspergillus* PCR was not used. Moreover, according to Mikulska et al., its sensitivity in COVID-19 ICU patients is rather low (40%) [39]. Previous research shows that serological tests detecting *Aspergillus* GM are also of little value in non-neutropenic patients because the circulating antigen is rapidly cleared by neutrophils [27]. Furthermore, the infection develops differently, not as aggressively as in patients with severe immunosuppression. The fungus may multiply more slowly only in the airways and infiltrate the blood vessels later than in the case of the classic forms of invasive pulmonary aspergillosis in immunocompromised patients. This means that blood markers of infection are not detected at the beginning, and their presence indicates a very advanced disease that may be difficult to treat despite the use of appropriate therapy. Testing for antigens in respiratory specimens is strongly recommended, as their presence in serum is extremely rare. The GM testing in BAL is characterized by high sensitivity (approx. 90%) and specificity (approx. 90%) and exceeds the specificity of serum GM testing (approx. 42%) and even culture methods (approx. 65%) [40,41]. Studies have shown that the detection of GM in BAL is a valuable tool in the diagnosis of aspergillosis in ICU patients; however, serial testing every 3–4 days are recommended, especially since the sensitivity of these tests is much lower in non-neutropenic patients compared to neutropenic patients [38,41]. ECMM/ISHAM CAPA guidance recommends for patients with SARS-CoV-2 screening testing for serum GM three times per week until discharge from ICU or defervescence for longer than 7 days with improved lung function and also testing of samples from the lower respiratory tract (not only BAL samples but also non-bronchoscopic materials) at least once a week [17]. During our analysis, we observed that one of the fundamental problems in the diagnosis of deep mycosis, including CAPA, in the ICU in our hospital is the lack of or too rare fungal antigen testing. GM assay concerned approximately 2% of patients hospitalized in ICU in the analyzed period, while 13% underwent fungal culture from lower respiratory samples. Moreover, the tests were usually performed only once, which makes their diagnostic value insignificant.

It should also be emphasized that the detection of GM in samples other than serum and BAL is not in accordance with the test manufacturer’s recommendations; therefore, it is not commonly used in UHK and only in exceptional situations at the request of the physician. This had a direct impact on the frequency of GM tests performed in our unit during the COVID-19 pandemic because non-bronchoscopic materials rather than BAL samples were collected from the respiratory tract.

GM detection remains the fastest diagnostic tool in invasive aspergillosis today. Currently, as an alternative to ELISA assay, immunochromatographic tests are available, which are much faster and easier to perform, especially under ICU conditions. Available GM point-of-care tests (*Aspergillus*-specific lateral flow device—LFD, and *Aspergillus* galactomannan lateral flow assay—LFA) allow to obtain a result within one hour after collecting a clinical sample, and their usefulness in the diagnosis of aspergillosis in ICU patients has already been confirmed [40,42,43,44,45]. Rapid antigen tests are not used in UHK for unknown reasons.

The analysis showed the lack of diagnostic regimens for invasive mycoses in our temporary COVID-19 intensive care ward. It resulted in the selective use of available diagnostic methods, and the performed tests do not provide clear information about the potential fungal etiological factor. During the analysis, no proven CAPA could be diagnosed, and we were not able to clearly determine the importance of *Aspergillus* strains cultured from the lower respiratory tract and positive GM results. Positive serum BDG results obtained in the analyzed period indicated fungal infection but not necessarily aspergillosis. This antigen is a component of the cell wall of many fungi, e.g., *Candida* sp. and *Pneumocystis* sp. *C. albicans* strains were isolated from all respiratory samples; however, their role in pneumonia is currently not recognized. In one case, *C. albicans* was isolated from blood culture with simultaneous positive results of GM and BDG in serum and negative culture of *Aspergillus* from the respiratory tract. The other mycological blood cultures were negative. It should be emphasized that this does not exclude fungal infection because only for *Candida* does the sensitivity of blood culture range from 50 to 75% or even lower [46]. Moreover, *Pneumocystis jirovecii*, a non-cultivable fungus that does not respond to antifungal treatment, could be the etiological factor of putative COVID-19 co-infection [47,48]. Unfortunately, the diagnosis of pneumocystosis was not performed in any patient hospitalized in the ICU in the analyzed period.

Too rarely used GM testing, limitations in detecting GM in non-bronchoscopic samples, as well as the lack of quick immunochromatographic GM assays, direct preparations from lower respiratory samples, and *Aspergillus* PCR testing might be the reasons for lower CAPA identification, but also prolonged diagnosis, delayed therapy, and high CAPA fatality rate, estimated at 76.5%. A correct diagnosis made at the right time is crucial for initiating appropriate and effective therapy. In patients with CAPA, *Aspergillus* isolation was obtained about one week stay in ICU, and the diagnosis was usually made after about two weeks of ICU hospitalization [22,49]. In our study, the median of days from the start of hospitalization to the final positive result of *Aspergillus* culture was 11 days (range 4–31 days). In 5 cases (29.4%), the results of the culture, which was the only diagnostic method used for mycological testing, were obtained after the patient’s death. The available serological methods were not used, and empirical antifungal therapy was not implemented despite the patient’s deteriorating condition and the previously used antibiotic therapy. It should be emphasized that all investigated patients presented predisposing factors for invasive mycoses, underwent invasive mechanical ventilation and systemic steroid therapy, and almost all received broad-spectrum antibiotics due to suspected bacterial co-infection, which were applicated before sample collection for mycological studies. Despite the lack of effectiveness of antibacterial therapy, empirical antifungal therapy was not administered, with one exception being when fluconazole was given after lower respiratory sample collection but before culture results and fungal antigens testings. The initiation of proper antifungal treatment is very important when aspergillosis is presumed. According to ECMM/ISHAM guidelines, recommended first-line treatment for invasive aspergillosis is intravenous voriconazole or isavuconazole [17]. In our study group, antifungal drugs were used only in 9 cases (52.9%), with voriconazole used in 7 and fluconazole, which could not act against *Aspergillus*, in 2. Antifungal susceptibility testing was performed for 12 *Aspergillus* strains, and none were resistant to voriconazole. Four strains were not tested, and the patients died, but two of them did not receive any antifungal treatment, and two had fluconazole, which is generally inactive against molds. The fatality rate in our research is up to 76.5% and is higher than that reported in the literature. A comprehensive systematic review and meta-analysis found that critically ill patients with CAPA have severely increased mortality, with a median of 54% (IQR 36–70%) in the suspected CAPA group [50]. In various studies, the mortality ranged from 44.5 to 66.7% to even 100% [49,51]. Despite the implementation of antifungal therapy in some patients, it was ineffective in most cases. This may have resulted from too late treatment or inappropriate therapy.

The main limitation of the reliable assessment of the epidemiology of CAPA in this study was the inability to identify proven CAPA cases and the low rate of probable CAPA cases. This is related to the type of clinical samples collected for mycological tests, which were predominantly non-bronchoscopic materials and affected the interpretation of the results and the classification of CAPA cases. The retrospective nature of the study does not allow for obtaining all the data and clarifying all doubtful issues.

## 5. Conclusions

The diagnosis of CAPA is a challenge for clinicians, especially in the ICU and for patients with other primary lung infections. There is a great need to implement more effective mycological diagnostics in the ICU and to increase the awareness of medical staff about the occurrence of invasive mycoses as a co-infection in the course of coronavirus disease. In our hospital, the available diagnostic methods should be used more effectively, i.e., serial testing of fungal antigens instead of single determinations, and also in non-bronchoscopic lower respiratory samples. Direct microscopic examinations of respiratory samples should be implemented, as well as additional diagnostic tests (immunochromatographic point-of-care GM assays and the PCR method) should be considered.

## Figures and Tables

**Figure 1 jof-09-00666-f001:**
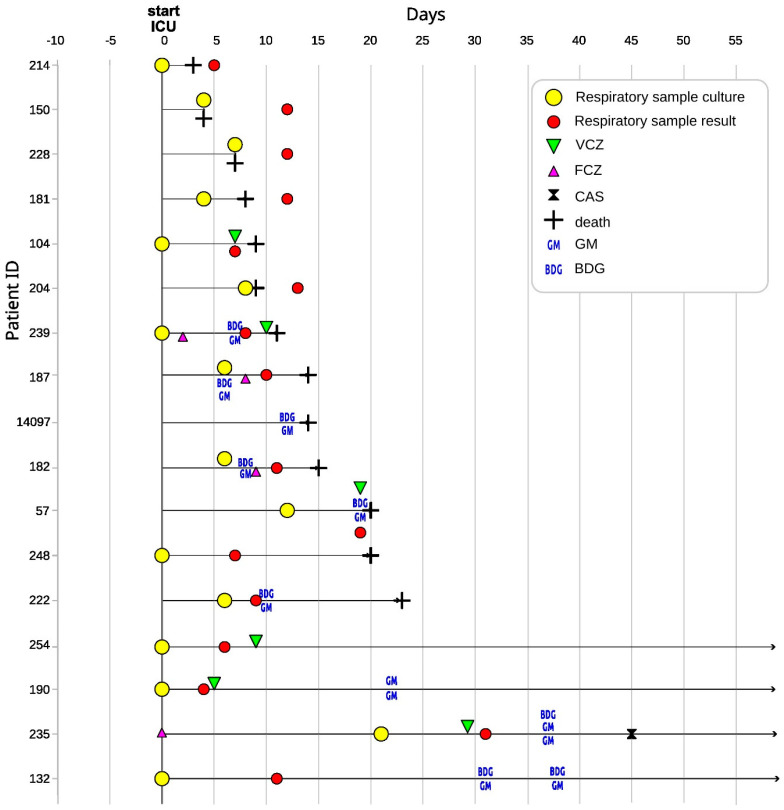
Timeline of mycological diagnostics in CAPA patients from admission to the intensive care unit. The figure template was taken from Lackner et al., 2022 [21]. ICU—intensive care unit, GM—galactomannan, BDG—1-3-β-D-glucan, VCZ—voriconazole, FCZ—fluconazole, CAS—caspofungin.

**Table 1 jof-09-00666-t001:** Diagnostic criteria used for analysis of aspergillosis cases.

Patient Number	Mycological Results							BM-AspICU Algorithm [16]					2020 ECMM/ISHAM Criteria for CAPA [17]		
	Positive Aspergillus Culture in BAL	Positive Aspergillus Culture in NBL	Positive Direct Examination of BAL or NBL Showing Hyphae	BAL Galactomannan	NBL Galactomannan	Serum/Plasma Galactomannan	Aspergillus qPCR								
								Host Factor	Risk Factor	Clinical Features	Radiological Features	IPA Definition	Host Factor	Clinical Features	CAPA Definition
57		X							X	X	X	PAoAC	X	X	Possible
104		X							X	X	X	PAoAC	X	X	Possible
132	X								X	X	X	PAoAC	X	X	Probable
150		X							X	X	X	PAoAC	X	X	Possible
181		X							X	X	X	PAoAC	X	X	Possible
182		X							X	X	X	PAoAC	X	X	Possible
187		X							X	X	X	PAoAC	X	X	Possible
190	X			X (index 0.85)					X	X	X	Probable	X	X	Probable
204		X						X	X	X	X	Probable	X	X	Possible
214		X							X	X	X	PAoAC	X	X	Possible
222		X							X	X	X	PAoAC	X	X	Possible
228		X							X	X	X	PAoAC	X	X	Possible
235		X			X (index 3.85)				X	X	X	PAoAC	X	X	Possible
239		X				X (index 7.60)		X	X	X	X	Probable	X	X	Probable
248		X							X	X	X	PAoAC	X	X	Possible
254		X							X	X	X	PAoAC	X	X	Possible
14097						X (index 8.48)		X	X	X	X	Probable	X	X	Probable

BAL—bronchoalveolar lavage, NBAL—non-bronchoscopic lavage, ECMM—European Confederation for Medical Mycology, ISHAM—International Society for Human and Animal Mycology, CAPA—COVID-associated pulmonary aspergillosis, IPA—invasive pulmonary aspergillosis, PAoAC—possible aspergillosis or *Aspergillus* colonization.

**Table 2 jof-09-00666-t002:** Factors associated with CAPA mortality.

Variable	CAPA			
	Total *n* = 17	Survivor *n* = 4	Non-Survivor *n* = 13	Statistics (*p*-Value)
**General host factors**				
Age median/range (years)	65/33–78	61/56–61	65/33–78	0.3052 ^U^
Female sex	41%	25%	46%	0.6029 ^F^
**Comorbidities**				
Diabetes	18%	25%	15%	1.0000 ^F^
Heart diseases	53%	50%	54%	1.0000 ^F^
Pulmonary diseases	12%	*n* = 0	15%	1.0000 ^F^
Malignancies	12%	*n* = 0	15%	1.0000 ^F^
Autoimmune diseases	12%	*n* = 0	15%	1.0000 ^F^
**Clinical condition and management prior to CAPA diagnosis**				
APACHE II	25/9–47	17/9–27	26/11–47	0.1254 ^U^
SAPS II	56/29–98	40/29–61	59/29–98	0.1123 ^U^
Mechanical ventilation	100%	100%	100%	
Lower respiratory tract bacterial infection or colonization	76%	25%	92%	0.2189 ^F^
Bacteremia	71%	50%	77%	1.0000 ^F^
Corticosteroids	100%	100%	100%	
Antibiotics	94%	75%	100%	0.2353 ^F^
**Mycological diagnostics of CAPA cases**				
Aspergillus culture	94%	100%	92%	1.0000 ^F^
*Aspergillus fumigatus*	87%	100%	83%	1.0000 ^F^
*Aspergillus niger*	12%	0	17%	1.0000 ^F^
Serum GM	53%	75%	46%	0.5765 ^F^
Index > 0.5	22%	0	33%	1.0000 ^F^
BAL GM	6%	25%	0	0.2353 ^F^
Index ≥ 1.0	0	0	0	
**2020 ECMM/ISHAM CAPA classification**				
Probable CAPA	23%	50%	15%	0.2189 ^F^
Possible CAPA	76%	50%	85%	0.2189 ^F^
**MB-AspICU IPA classification**				
Probable IPA	23%	25%	23%	1.0000 ^F^
Possible IPA or *Aspergillus* colonization	76%	75%	77%	1.0000 ^F^
**CAPA treatment**				
VCZ	41%	75%	31%	1.0000 ^F^
CAS	6%	25%	0	0.2353 ^F^
AMB	0	0	0	
2020 ECMM/ISHAM Probable CAPA + VCZ	12%	25%	8%	0.4265 ^F^
2020 ECMM/ISHAM Possible CAPA + VCZ	29%	25%	31%	0.5378 ^F^
BM-AspICU Probable IPA + VCZ	12%	25%	8%	0.4265 ^F^
BM-AspICU Possible IPA or Aspergillus colonization + VCZ	29%	25%	31%	0.5378 ^F^

The table template was taken from Calderón-Parra et al., 2022 [20]. The cut-off values for GM were adopted from ECMM/ISHAM CAPA classification [17]. ICU—intensive care unit, GM—galactomannan, BAL—bronchoalveolar lavage, ECMM—European Confederation for Medical Mycology, ISHAM—International Society for Human and Animal Mycology, CAPA—COVID-associated pulmonary aspergillosis, IPA—invasive pulmonary aspergillosis, VCZ—voriconazole, CAS—caspofungin, AMB—amphotericin B, U—Mann–Whitney U Test, F—Fisher’s Exact Test.

## Data Availability

The datasets generated during and/or analyzed during the current study are not publicly available but are available from the corresponding author upon reasonable request.

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
