# Peer review of "COVID-19-Associated Pulmonary Aspergillosis in Intensive Care Unit Patients from Poland"

_jof, 2023, doi:10.3390/jof9060666_

Round 1

Reviewer 1 Report

In the present manuscript, the authors assess the incidence of CAPA among ICU patients in Poland and analyse applied diagnostic and therapeutic procedures. This is a well-written paper, the results are clearly presented. The discussion and conclusions are in accordance with the results. Appropriate citations and references are used. The methodology used needs improvement and the authors are advised to make these ammendments before considering for publication. In particular,

-         - The authors are advised to refer to and integrate accordingly in their analysis the recently proposed BM-AspICU algorithm (Hamam J, Navellou JC, Bellanger AP, Bretagne S, Winiszewski H, Scherer E, Piton G, Millon L; Collaborative RESSIF group. New clinical algorithm including fungal biomarkers to better diagnose probable invasive pulmonary aspergillosis in ICU. Ann Intensive Care. 2021 Mar 8;11(1):41. doi: 10.1186/s13613-021-00827-3. PMID: 33683480; PMCID: PMC7). This a clinical algoritm takes into account fungal biomarkers, such as the GM antigen and the Aspergillus qPCR for the classification.

-         - In Table 1, factors associated with CAPA occurrence and outcome are presented.  Fisher’s exact test should be applied to determine whether or not there is a significant association.

-        -  Lines 154-158: The authors present the therapeutic administration of antibacterial drugs, but they do not present the bacterial pathogens recovered from these patients.

Author Response

Dear Sir or Madam,

I am submitting a revised version of our article "COVID-19-associated pulmonary aspergillosis in intensive care unit patients from Poland" by Magdalena Skóra, Mateusz Gajda, Magdalena NamysÅ‚, Jerzy Wordliczek, Joanna Zorska, Piotr PiekieÅ‚ko, Barbara Å»óÅ‚towska, PaweÅ‚ KrzyÅ›ciak, Piotr B. Heczko, and Jadwiga Wójkowska-Mach (Mnuscript ID: jof-2412694) with a request for publication in Journal of Fungi Special Issue "Young Investigator in Fungal Infections 2.0".

We are very grateful for a careful revision of our paper. We have taken onboard the Reviewer’s suggestions and we did our best to correct and modify the manuscript accordingly. Changes made to the manuscript are described below.

In accordance with the recommendations of Reviewer 2, the title of the article was changed. The manuscript text, the contents of the tables and the titles of the tables were modified, due to Reviewers recommendations.

Reviewer:

The authors are advised to refer to and integrate accordingly in their analysis the recently proposed BM-AspICU algorithm (Hamam J, Navellou JC, Bellanger AP, Bretagne S, Winiszewski H, Scherer E, Piton G, Millon L; Collaborative RESSIF group. New clinical algorithm including fungal biomarkers to better diagnose probable invasive pulmonary aspergillosis in ICU. Ann Intensive Care. 2021 Mar 8;11(1):41. doi: 10.1186/s13613-021-00827-3. PMID: 33683480; PMCID: PMC7). This a clinical algoritm takes into account fungal biomarkers, such as the GM antigen and the Aspergillus qPCR for the classification.

Response

According to the advice we analysed aspergillosis cases also in accordance to BM-AspICU. A summary of the analysis is included in a newly created table which has been incorporated into the manuscript as Table 1. The detailed information on clinical signs, host factors and radiological changes can be found in the modified Table S1.

Reviewer:

In Table 1, factors associated with CAPA occurrence and outcome are presented.  Fisher’s exact test should be applied to determine whether or not there is a significant association.

Response:

Reviewer's comment has been considered. Statistical analysis was performed, and the results are shown in the table (Table 2). A new table (now Table 1) has been added to the text, therefore the numbering of the tables has changed. The title of the old Table 1 (now Table 2) have been modified according to Reviewer 2 recommendations.

Reviewer:

Lines 154-158: The authors present the therapeutic administration of antibacterial drugs, but they do not present the bacterial pathogens recovered from these patients.

Response:

Information on isolated bacterial pathogens has been added. The detailed data are provided in Table S2 in the Supplementary Information. A new column "Bacteria isolated from lower respiratory tract materials" was created. The title of the table has also been changed in order to more adequately name the data contained in the table. The information on bacterial co-infections or colonization is also included in Table 2.

Reviewer 2 Report

This is not a bad work; information on Invasive fungal infections is essential in medical mycology. However, the limitation when describing CAPA cases is to ensure the correct classification. 

-First, I suggest removing putative from the title; ISHAM criteria do not consider putative as a possibility, only proven, probable and possible. A better title, "COVID-19-associated pulmonary aspergillosis in intensive care unit patients from Poland"

-Classification of CAPA cases should be clearer; if use ISHAM cannot use putative if the AspICU criteria are not being used. Change this from the table title on supplementary material. It also considers the clinical criteria, some of its possible cases, which are the most difficult to classify, have simultaneous isolation as bacteremia or other infection. One of the clinical criteria is not having a justification for the worsening or lack of improvement. In addition, it would be helpful in the case table to clarify the clinical and radiological manifestations of the diagnosis. 

How many cases had tomography or xray for the CAPA diagnosis, and what were the findings? Not only can the mycological criteria be met, but the clinical and radiological criteria are also required. 

As there are few cases, supplementary table 1 could be part of the article with some changes. Add the clinical and radiological criteria to know if they are cases of CAPA or not. Remember that the isolation of Aspergillus in the respiratory tract can mean colonization. Hence the relevance of the criteria.

Regarding antifungal treatment, first should describe homogeneous, always antifungal, sometimes they write antimycotic then antifungal. Line 212-230 is very confusing; perhaps it would be better explained in the table. Explain why fluconazole since it is not an antimold. 

Table 1. Compare live CAPA vs. dead CAPA; this is a mortality analysis; not know the factors associated with CAPA; this could only be done if compared with controls without CAPA since the Table title requires a change.

In the table, they make a comparison, and we do not see either the OR or the p result of that statistical comparison, whether they have performed a Chi2 test or whatever corresponds.

The table mixes the patients' comorbidities with characteristics of the covid-19 episode, then covid treatment with CAPA treatment, specify and separate. If they used the template from Calderon-Parra et al., see how the authors display the results organized and the comparison p-value. You did not mention comparing mortality in your introduction or methods; I suggest adding it as a secondary objective.

Discussion

If the primary objective is to describe the incidence (rather is prevalence), your result is what they should discuss first. Then they talk about the classification; why do you think they have a low percentage of probable CAPA? They mention it later; you must order the discussion according to its main objective and then specify them. Discuss why the low prevalence is compared to what is reported locally for aspergillosis.

The conclusion is long and seems like a discussion. Write again and focus on what you found according to the main objective and final opinion regarding the need for a better diagnosis.

Author Response

Dear Sir or Madam,

I am submitting a revised version of our article "COVID-19-associated pulmonary aspergillosis in intensive care unit patients from Poland" by Magdalena Skóra, Mateusz Gajda, Magdalena NamysÅ‚, Jerzy Wordliczek, Joanna Zorska, Piotr PiekieÅ‚ko, Barbara Å»óÅ‚towska, PaweÅ‚ KrzyÅ›ciak, Piotr B. Heczko, and Jadwiga Wójkowska-Mach (Mnuscript ID: jof-2412694) with a request for publication in Journal of Fungi Special Issue "Young Investigator in Fungal Infections 2.0".

We are very grateful for a careful revision of our paper. We have taken onboard the Reviewer’s suggestions and we did our best to correct and modify the manuscript accordingly. Changes made to the manuscript are described below.

Reviewer: First, I suggest removing putative from the title; ISHAM criteria do not consider putative as a possibility, only proven, probable and possible. A better title, "COVID-19-associated pulmonary aspergillosis in intensive care unit patients from Poland"

Response: According to the Reviewer suggestion the title was modified.

Reviewer: Classification of CAPA cases should be clearer; if use ISHAM cannot use putative if the AspICU criteria are not being used. Change this from the table title on supplementary material. It also considers the clinical criteria, some of its possible cases, which are the most difficult to classify, have simultaneous isolation as bacteremia or other infection. One of the clinical criteria is not having a justification for the worsening or lack of improvement. In addition, it would be helpful in the case table to clarify the clinical and radiological manifestations of the diagnosis.

Response: The title of Table S1 in Supplementary Information was modified – the term „putative” was removed from the title. The clinical and radiological manifestations has been supplemented in the Table S1. Additional columns have been added to the table. The data were obtained from patients' medical documentation. As recommended by Reviewer 1, we also analyzed our cases using an algorithm BM-AspICU by Hamam et al. Therefore, we created a new table (now Table 1) in which we summarized the clinical, radiological and mycological evidences and defined cases according to the 2020 ECMM/ISHAM consensus criteria and the BM-AspICU algorithm.

Reviewer: How many cases had tomography or xray for the CAPA diagnosis, and what were the findings? Not only can the mycological criteria be met, but. the clinical and radiological criteria are also required

Response: The information on radiological findings and clinical manifestations were included in Table S1 (additional columns have been added to the table).

Reviewer: As there are few cases, supplementary table 1 could be part of the article with some changes. Add the clinical and radiological criteria to know if they are cases of CAPA or not. Remember that the isolation of Aspergillus in the respiratory tract can mean colonization. Hence the relevance of the criteria.

Response: The clinical and radiological findings were included in Table S1. We decided to leave the Table S1 in Supplementary Information and add a new table to the manuscript (now Table 1).

Reviewer: Regarding antifungal treatment, first should describe homogeneous, always antifungal, sometimes they write antimycotic then antifungal. Line 212-230 is very confusing; perhaps it would be better explained in the table. Explain why fluconazole since it is not an antimold.

Response: We standardized the vocabulary in accordance with the reviewer's recommendations (antifungal, not antimycotic). The data described in lines 212-230 are included in Supplementary Information Table S1 and Table S2. We are unable to clearly explain the administration of fluconazole. This drug was administered prior to positive Aspergillus culture from lower respiratory tract samples. These were individual medical decisions, impossible to explain retrospectively. In two cases, fluconazole was administered after a positive serum beta-D-glucan result, so perhaps this fact and the suspicion of invasive candidiasis were the reason for the administration.

Reviewer: Table 1. Compare live CAPA vs. dead CAPA; this is a mortality analysis; not know the factors associated with CAPA; this could only be done if compared with controls without CAPA since the Table title requires a change

Response: After addition of the new table to the manuscript Table 1 became Table 2. The title of the table and its content were changed.

Reviewer: In the table, they make a comparison, and we do not see either the OR or the p result of that statistical comparison, whether they have performed a Chi2 test or whatever corresponds.

Response: We performed statistical analysis. The results are included in Table 2 and the methodology was described in the section „Materials and Methods”.

Reviewer: The table mixes the patients' comorbidities with characteristics of the covid-19 episode, then covid treatment with CAPA treatment, specify and separate. If they used the template from Calderon-Parra et al., see how the authors display the results organized and the comparison p-value.

Response: We reorganized Table 2 (former Table 1). We included general host factors, comorbidities, ICU stay management prior to CAPA diagnosis, CAPA diagnosis and CAPA treatment. We added p-value to the table.

Reviewer: You did not mention comparing mortality in your introduction or methods; I suggest adding it as a secondary objective.

Response: We added the analysis of CAPA mortality as a secondary objective.

Reviewer: Discussion. If the primary objective is to describe the incidence (rather is prevalence), your result is what they should discuss first. Then they talk about the classification; why do you think they have a low percentage of probable CAPA? They mention it later; you must order the discussion according to its main objective and then specify them. Discuss why the low prevalence is compared to what is reported locally for aspergillosis.

Response: We ordered the discussion according to Reviewer comments.

Reviewer: The conclusion is long and seems like a discussion. Write again and focus on what you found according to the main objective and final opinion regarding the need for a better diagnosis.

Response: We have shortened the conclusions, leaving only those that result from our analysis.

Round 2

Reviewer 1 Report

We thank the authors for providing and improving the manuscript for publication in JoF.

Author Response

Dear Sir or Madam,

on behalf of all the authors, thank you for the reviews, which were extremely helpful in the preparation of the article.

Yours faithfully,

Magdalena Skóra

Reviewer 2 Report

Dear authors, thank you for accepting the suggestions and changes.

I have a few comments. 

  1. On lines 26, 29, 291, it still says the word " incidence". Remember that if you want to report an incidence rate, they must specify "x patients or cases per x person-time (year,month,etc)." 

 You can find an explanation in this CDC source. https://www.cdc.gov/csels/dsepd/ss1978/lesson3/section2.html#:~:text=Prevalence%20and%20incidence%20are%20frequently,during%20a%20particular%20time%20period.

  1. On line 466-468 regarding limitations, one of the most significant limitations, more than the retrospective nature, is that the highest percentage of CAPA cases are possible CAPA and not probable or proven. The limitation of having samples obtained with BAL may be associated with a misclassification bias. , which I know was mentioned above. Even so, mortality is high, which should be specified in your discussion, and the low prevalence in your city or country.

Author Response

Dear Sir or Madam,

on behalf of all the authors, thank you very much for your thorough evaluation and analysis of our article. We have tried to improve and supplement the text as suggested. Please find below a detailed description of the changes made. I hope that the modifications will meet the Reviewer's expectations.

Yours faithfully,

Magdalena Skóra

Reviewer:

On lines 26, 29, 291, it still says the word " incidence". Remember that if you want to report an incidence rate, they must specify "x patients or cases per x person-time (year,month,etc)."

You can find an explanation in this CDC source. https://www.cdc.gov/csels/dsepd/ss1978/lesson3/section2.html#:~:text=Prevalence%20and%20incidence%20are%20frequently,during%20a%20particular%20time%20period.

Response:

According to the Reviewer comment concerning the usage of terms „prevalence” and „incidence”, and according to the CDC publication recommended by the Reviewer (CDC Principles of Epidemiology in Public Health Practice. https://www.cdc.gov/csels/dsepd/ss1978/SS1978.pdf Accessed 6 June 2023), we removed the terms "prevalence" and "incidence" from the article and introduced terms „morbidity”, „incidence density rate per 10 000 patient days” and „incidence rate”. Changes in the text concern the lines: 26, 29, 30, 91, 181, 182, 297, 301-303, 536.

The methodology used (according to the above CDC publication) has been added to the Materials and Methods section and the reference has been added to the reference list (reference number 18). 

In the section Materials and Methods the information on patient-days and a new paragraph on the methodology of the epidemiological analysis were added (lines 99-100 and lines 124-126, respectively).

Reviewer:

On line 466-468 regarding limitations, one of the most significant limitations, more than the retrospective nature, is that the highest percentage of CAPA cases are possible CAPA and not probable or proven. The limitation of having samples obtained with BAL may be associated with a misclassification bias. , which I know was mentioned above.

Response:

Reviewer suggestions have been included in the text. We have modified the last paragraph of the "Discussion" section concerning limitations (lines 568-574).

Reviewer:

Even so, mortality is high, which should be specified in your discussion, and the low prevalence in your city or country.

Response:

Due to Reviewer comment we referred to the low prevalence of CAPA and high mortality and the likely reasons for these results. In the "Discussion" section, second paragraph, lines 307-316 we added and modified the text.